# Shrubs Should Be Valued: The Functional Traits of *Lonicera fragrantissima* var. *lancifolia* in a Qinling Huangguan Forest Dynamics Plot, China

**Anxia Han, Jing Qiu, Ruoming Cao, Shihong Jia, Zhanqing Hao and Qiulong Yin \***

School of Ecology and Environment, Northwestern Polytechnical University, Xi'an 710129, China; a15176938636@163.com (A.H.); qiuqiu18536062981@163.com (J.Q.); cao@nwpu.edu.cn (R.C.); shihong.jia@nwpu.edu.cn (S.J.); zqhao@nwpu.edu.cn (Z.H.)
* Correspondence: yinql@nwpu.edu.cn

**Abstract:** Previous studies have focused on the functional traits of trees, while undergrowth shrubs have not received the same attention. We collected 97 shrubs from 6 habitats in 3 diameter classes to measure the functional traits of *Lonicera fragrantissima* var. *lancifolia*, which is one of the dominant species in the shrub layer of the Qinling Huangguan plot. We found that leaf thickness (LT) decreased with an increase in diameter classes. Other functional traits did not change significantly with the diameter classes. Most of the functional traits changed with the habitats, which may be influenced by topography and soil. On the whole, *Lonicera fragrantissima* var. *lancifolia* showed low variation, which indicates that its growth was stable and good. The relationships between functional traits within species was in accordance with the leaf economic spectrum. The positive correlation between soil total nitrogen (STN) and C:N verified the "nutrition luxury hypothesis".

**Keywords:** functional traits; diameter class; micro-habitat; soil

## 1. Introduction

Plant functional traits are a series of indicators that characterize the growth state of plants, which has an important effect on plants and ecosystems and can reflect the choice of plant survival strategy [1–3]. The relationship between an environment and plants can be explored through the variations in functional traits since they respond to environmental conditions. The variations in plant functional traits are affected by plant sizes, climatic factors, geographical conditions, soil factors, and so on [4].

Functional traits are coordinated or weighed with each other [5], forming the leaf economic spectrum (LES). The LES has been put forward by Wright et al. in 2004 [3], dividing plants into two types: quick investment-return species and slow investment-return species. At present, the LES has been extended to the whole plant, including root traits, stem traits, and plant size. Studies have shown that intraspecific traits of tree species vary greatly [6,7] and are systematically dependent on individual development [8]. For example, leaf mass per unit area (LMA) of canopy trees significantly varied with growth stage and/or tree height [9,10]. Generally, the growth of plants decreases the leaf area (LA) and specific leaf area [11,12], but increases the leaf dry matter content (LDMC) [13]. These size-dependent trait changes are a result of the adaptation and plasticity of plants, which adopt more conservative light- and water-use strategies under strong light and drought stresses [14]. However, some studies have shown that large individuals of undergrowth species tend to have larger leaves than small individuals [15,16], which indicates a freer strategy of resource utilization.

The effects of environmental factors on functional traits are diverse in different spatial scales. On a global scale, functional traits are affected by abiotic factors, mainly climate [17]. At a landscape scale, human-induced disturbances are the main factors [18]. At a local scale,

the main influencing factors are topography and soil [19,20]. The relationships between environment and functional traits have been emphasized on a large scale [3,21]. However, little is known about their relationships at a local scale. On a small scale, the variations in topography and soil can lead to differences in habitat environments, such as water availability, air pressure, temperature, and light. Kizawa et al. and Poorter et al. found that specific leaf area became smaller under high light and low water availability and soil nutrients [22]. Kühn et al. found that the changes in intraspecific leaf traits for native and non-native species showed different patterns along altitudinal gradients [23,24].

Our research on the functional traits of *Lonicera fragrantissima* var. *lancifolia* in the Huangguan study plot of Qinling Mountains belongs to local-scale research. The study plot is located in Qinling Mountains. Qinling Mountains, the boundary between North and South China, have a typical geographical location. It is the transition zone between a subtropical humid climate and warm temperate continental climate, characterized by moderate rain and temperature and high species diversity. This area is one of the centers of distribution of *L. fragrantissima* var. *lancifolia* in China. *Lonicera fragrantissima* var. *lancifolia* can produce chlorogenic acid. Its fruit is edible and has potential development value [25]. *Lonicera fragrantissima* var. *lancifolia* plays an important role in the material circulation of ecosystems, vegetation renewal, and biodiversity maintenance. However, most studies have focused on its propagation technology, application value, and spatial patterns [26,27]. We are still far from comprehending the functional traits of *L. fragrantissima* var. *lancifolia*. Understanding of its functional traits cannot only guide its cultivation and predict the impact of human disturbance on the ecosystem, but also provide support for the theory of the economic spectrum of intraspecific leaves. Therefore, we analyzed its functional traits under microenvironment conditions in a study plot on Qinling Mountains. We hypothesized that (1) from a small tree to a big tree, *L. fragrantissima* var. *lancifolia* will transition from a "fast" strategy to a "slow" strategy; (2) the correlation of intraspecific traits is consistent with the LES; and (3) elevation gradients in a habitat have a great influence on traits.

## 2. Materials and Methods

### 2.1. Study Site

A study plot of 25 ha (500 m × 500 m) was established on Qinling Mountains (33°32′20.61″ N, 108°22′25.62″ E, 1280.3–1581.8 m a.s.l.) in 2019, which is called the Qinling Huangguan plot. The Qinling Huangguan plot is located in Changqing National Nature Reserve, China. The climate is subtropical humid, with an average annual rainfall of 908.0 mm and an average annual temperature of 12.3 °C [28]. The plot has been divided into 625 small sub-quadrats of 20 m × 20 m, in which trees with a DBH ≥ 1 cm were numbered and identified. *Lonicera fragrantissima* var. *lancifolia* is the most dominant shrub in the plot, with 2611 plants. It is a subspecies of *Lonicera fragrantissima* of the *Lonicera* Linn. genus, in the Caprifoliaceae Juss. family. It is a deciduous shrub with an adult tree height of 2 m. According to the actual growth situation and survey data in the plot, *L. fragrantissima* var. *lancifolia* was divided into three diameter classes, namely, 1 cm ≤ small shrubs (S) < 1.5 cm, 1.5 cm ≤ medium shrubs (M) < 2 cm, and 2 cm ≤ large shrubs (L).

### 2.2. Habitat Classification

Because of the differing altitudes, slopes, convexities, and aspects of the Qinling Huangguan plot, Ward hierarchical clustering analysis was adopted [29]. Combined with the actual topographic conditions, the Qinling Huangguan plot was divided into six topographic habitats: valley, low-ridge, slope, gully, high-ridge, and terrace (Table 1).

**Table 1.** Habitat classification.

| Habitat Abbreviations | Name | Quadrat Number | Total Area (ha) | Mean Elevation (m) | Mean Slope (°) | Mean Convexity (m) | Mean Aspect (°) |
|---|---|---|---|---|---|---|---|
| VA | valley | 39 | 1.56 | 1322.35 ± 2.40 | 10.93 ± 0.51 | −2.08 ± 0.44 | 317.70 ± 6.77 |
| LR | low-ridge | 136 | 5.44 | 1360.76 ± 2.43 | 24.78 ± 0.55 | 1.00 ± 0.28 | 45.29 ± 6.51 |
| SL | slope | 154 | 6.16 | 1374.92 ± 4.12 | 27.13 ± 0.45 | −1.01 ± 0.17 | 327.39 ± 3.53 |
| GU | gully | 145 | 5.80 | 1442.06 ± 3.63 | 36.20 ± 0.40 | −0.89 ± 0.19 | 20.11 ± 1.19 |
| HR | high-ridge | 59 | 2.36 | 1503.84 ± 4.38 | 31.69 ± 0.57 | 1.15 ± 0.24 | 344.73 ± 1.94 |
| TE | terrace | 92 | 3.68 | 1526.10 ± 2.87 | 26.00 ± 0.56 | 2.40 ± 0.37 | 40.34 ± 8.37 |

*2.3. Measurements of Soil Factors*

The soil samples were collected at a depth of 10 cm, with a total of 972 sampling points in the plot. Soil total phosphorus (STP) was determined by perchloric acid-sulfuric acid-molybdenum antimony anti-colorimetry. Soil total nitrogen (STN) was determined by the Kjeldahl method. The pH was measured by the potential method, and the water-soil ratio was 2.5:1. The soil indexes around each tree/shrub were calculated by the ordinary Kriging interpolation method.

*2.4. Plant Sampling and Trait Measurements*

In total, 3–7 individuals of large, medium, and small shrubs were selected from each habitat, respectively. A total of 97 shrubs were collected, and the minimum distance between each shrub was 1 m (Figure 1). The leaves and annual or perennial branches (3 replicates) from each shrub were collected. First, the leaf thickness (LT) of complete leaves (3 replicates) was immediately measured by a Vernier caliper in the field. Three positions from top to bottom of the sampling leaf were chosen to avoid the primary and secondary veins. The average LT of a sampling shrub was estimated by 9 replicates in total. The fresh weight was weighed with a balance. We scanned the whole leaf and calculated the LA with Motic Images Plus 2.0 software (Xiamen, Fujian, China). All samples were collected within 15 days in order to avoid the influence of time heterogeneity on leaf traits [13]. Then, the samples of branches and leaves were dried at 60 °C for 72 h. WD was estimated by the upper and lower diameters, lengths, and dry weights of branches. Leaf carbon content (LCC), LNC, and LPC were measured with a Euro Vector EA3000. LDMC was calculated as the ratio of dry weight to fresh weight. LMA was calculated as the ratio of leaf dry weight to LA. Leaf tissue density (LD) was the ratio of LMA to LT.

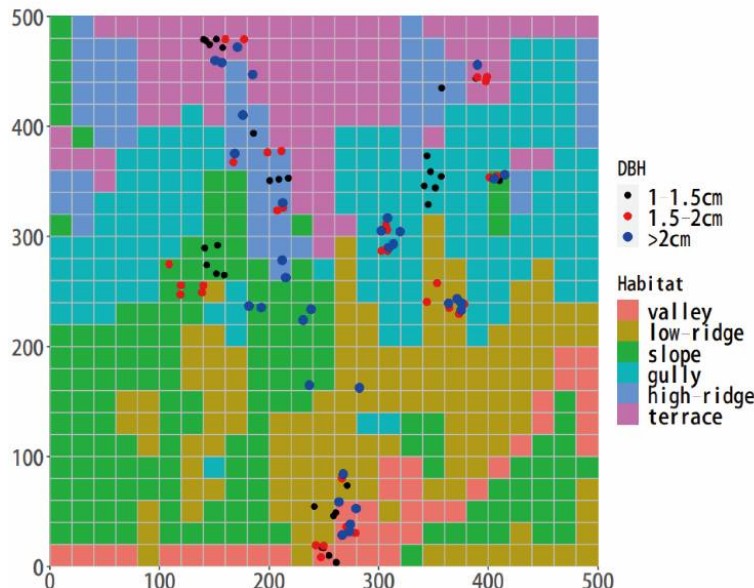

**Figure 1.** Distribution of sampling sites of *L. fragrantissima* var. *lancifolia* in the Qinling Huangguan plot.

### 2.5. Statistical Analysis

Simple linear regression with 95% confidence intervals was used to analyze the correlation between the functional traits and DBH [15,30]. The coefficient of variation (CV) was used to show the variability in each functional trait (CV = standard deviation/mean). The Pearson correlation coefficient was used to calculate the correlations among the functional traits, and canonical correspondence analysis (CCA) was used to analyze the relationships between the functional traits and influencing factors.

Excel was used for basic data processing, and R 4.2.0 (R Core Team, Vienna, Austria, 2022) was used for data analysis and creation of figures.

## 3. Results

### 3.1. Variations of 11 Functional Traits with Shrub Sizes and Habitats

LT, LMA, and C:N decreased with an increase in diameter classes, while LDMC, LA, TD, WD, LNC, LCC, LPC, and N:P increased with an increase in diameter classes (Table 2). LT significantly decreased with an increase in diameter classes. There were no significant relationships between the other functional traits and diameter classes (Table 2).

**Table 2.** Changes in functional traits of *L. fragrantissima* var. *lancifolia* with shrub sizes. Trait abbreviations: LT (leaf thickness); LDMC (leaf dry matter content); LA (leaf area); LMA (leaf mass per unit area); TD (leaf tissue density); WD (wood density); LNC (leaf nitrogen content); LCC (leaf carbon content); LPC (leaf phosphorus content).

| Relationship (y–x) | Regression Equation | $p$ | $R^2$ |
|---|---|---|---|
| LT–DBH | $y = -0.0046x + 0.13$ | 0.028 | 0.050 |
| LDMC–DBH | $y = 0.0030x + 0.36$ | 0.57 | 0.0035 |
| LA–DBH | $y = 0.71x + 19$ | 0.24 | 0.015 |
| LMA–DBH | $y = -8.4 \times 10^{-5}x + 0.0051$ | 0.44 | 0.0062 |
| TD–DBH | $y = 0.00096x + 0.040$ | 0.34 | 0.0095 |
| WD–DBH | $y = 0.030x + 0.63$ | 0.22 | 0.016 |
| LNC–DBH | $y = 0.36x + 18$ | 0.16 | 0.021 |
| LCC–DBH | $y = 1.8x + 458$ | 0.23 | 0.015 |
| LPC–DBH | $y = 0.026x + 2.7$ | 0.74 | 0.0011 |
| C:N–DBH | $y = -0.31x + 25$ | 0.35 | 0.0094 |
| N:P–DBH | $y = 0.085x + 7.1$ | 0.73 | 0.0013 |

There were different variations in functional traits across habitats. LT was the highest in high-ridge and terrace. LDMC was the highest in low-ridge, followed by valley, high-ridge, and terrace, and the lowest in slope and gully. LA was the highest in slope and the lowest in terrace. LMA was the highest in high-ridge and the lowest in valley. TD was the highest in low-ridge and slope, and the lowest in valley. WD was the highest in valley and terrace, and the lowest in gully. LNC was the highest in gully and the lowest in terrace. There was no significant difference in LCC among the different habitats. LPC was the highest in valley. C:N was the highest in terrace and the lowest in gully. There was no significant difference in N:P for slope, gully, high-ridge, and terrace, whereas the lowest was in valley (Figure 2).

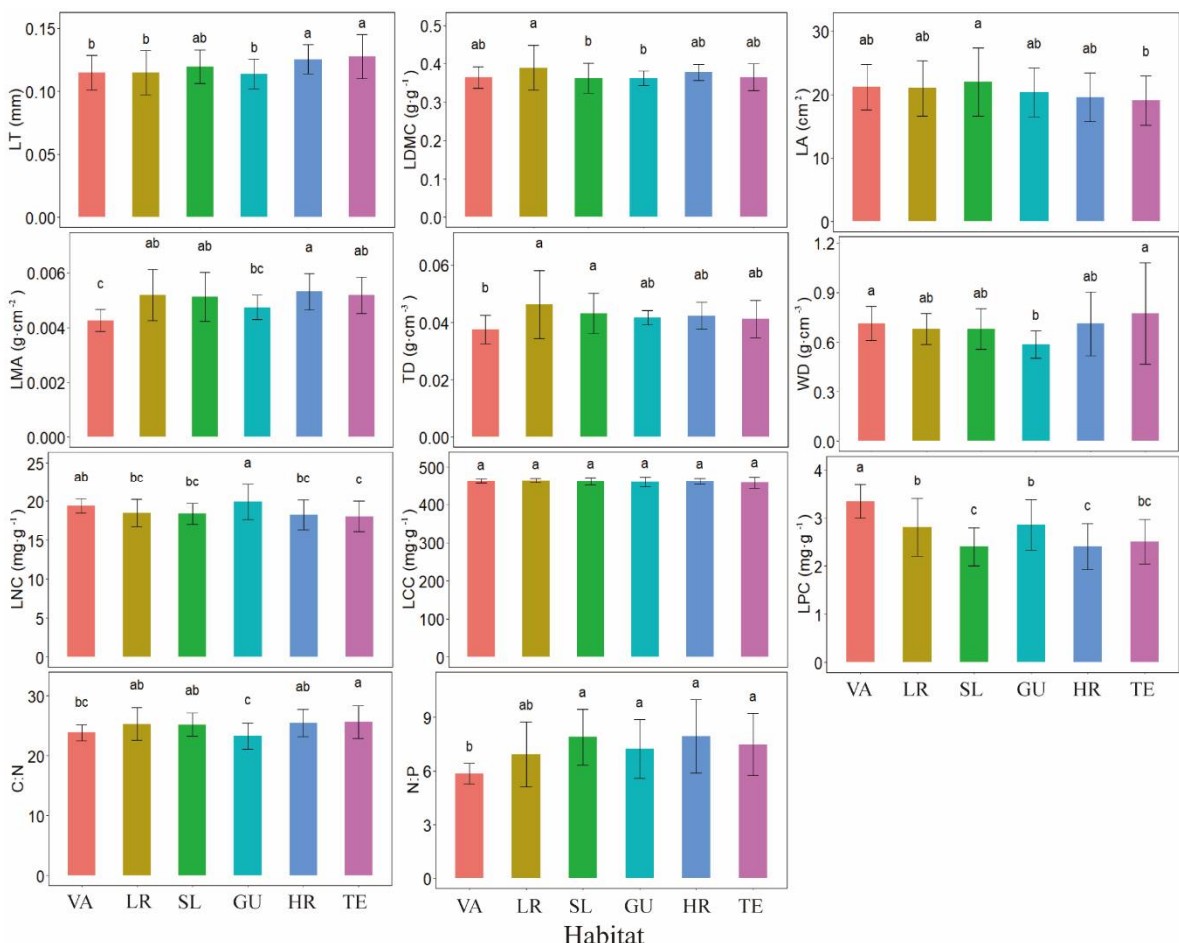

**Figure 2.** Changes in functional traits of *L. fragrantissima* var. *lancifolia* across habitats. Error bars represent the standard error. a is the largest, followed by b, and c is the smallest. Different letters denote significant differences among different habitats at *p* < 0.05. Habitat abbreviations: VA (valley); LR (low-ridge); SL (slope); GU (gully); HR (high-ridge); TE (terrace). Trait abbreviations: LT (leaf thickness); LDMC (leaf dry matter content); LA (leaf area); LMA (leaf mass per unit area); TD (leaf tissue density); WD (wood density); LNC (leaf nitrogen content); LCC (leaf carbon content); LPC (leaf phosphorus content).

### 3.2. Variability of Functional Traits in Different Diameter Classes

Except for the moderate variations in LA in small and medium shrubs, WD in small and large shrubs, LPC in medium and large shrubs, and N:P in small, medium, and large shrubs, all the other traits showed low variation. No traits showed strong variation. WD had the largest coefficient of variation and LCC had the smallest coefficient of variation (Table 3).

**Table 3.** Variability of the functional traits in different diameter classes.

| Diameter Class | Coefficient of Variation of Functional Traits/% | | | | | | | | | | |
|---|---|---|---|---|---|---|---|---|---|---|---|
| | LT | LDMC | LA | LMA | TD | WD | LNC | LCC | LPC | C:N | N:P |
| S | 9.8 | 10.1 | 20.46 | 18.22 | 16.49 | 20.39 | 7.32 | 1.91 | 19.51 | 7.12 | 20.62 |
| M | 13.53 | 7.01 | 23.64 | 11.96 | 11.71 | 19.41 | 10.15 | 2.87 | 21.03 | 9.77 | 27.46 |
| L | 13.66 | 11.73 | 17.52 | 16.8 | 19.81 | 32.9 | 11.46 | 1.99 | 21.91 | 11.36 | 22.64 |

Diameter class abbreviations: S (small shrub), M (medium shrub), and L (large shrub). A CV ≤ 0.2 is low variation, 0.2 < CV ≤ 0.5 is medium variation, and a CV > 0.5 is high variation. Trait abbreviations: LT (leaf thickness); LDMC (leaf dry matter content); LA (leaf

area); LMA (leaf mass per unit area); TD (leaf tissue density); WD (wood density); LNC (leaf nitrogen content); LCC (leaf carbon content); LPC (leaf phosphorus content).

### 3.3. Correlations among Functional Traits

LT had a significantly positive correlation with LMA and correlated negatively with TD. LDMC had significantly positive correlations with five functional traits (LMA, TD, WD, LCC, and C:N). LA was positively correlated with LNC but negatively correlated with C:N ($p < 0.01$). LMA had a significantly positive correlation with six functional traits (LT, LDMC, TD, WD, LCC, and C:N). LNC was positively correlated with three traits (LA, LCC, and N:P) and negatively correlated with two traits (WD and C:N). LCC had a significantly positive correlation with four functional traits (LDMC, LMA, TD, and LNC). LPC had a significantly negative correlation with N:P. C:N had the strongest correlations with the other traits (Table 4).

**Table 4.** Pearson correlation coefficient of 11 functional traits of *L. fragrantissima* var. *lancifolia*.

|      | LT | LDMC | LA | LMA | TD | WD | LNC | LCC | LPC | C:N |
|------|------|------|------|------|------|------|------|------|------|------|
| LDMC | −0.12 | | | | | | | | | |
| LA | 0.14 | −0.17 | | | | | | | | |
| LMA | **0.31 \*\*** | **0.76 \*\*** | −0.02 | | | | | | | |
| TD | **−0.44 \*\*** | **0.82 \*\*** | −0.14 | **0.70 \*\*** | | | | | | |
| WD | 0.05 | **0.26 \*** | −0.08 | **0.24 \*** | 0.18 | | | | | |
| LNC | −0.04 | −0.16 | **0.28 \*\*** | −0.18 | −0.16 | **−0.33 \*\*** | | | | |
| LCC | −0.17 | **0.42 \*\*** | 0.03 | **0.32 \*\*** | **0.38 \*\*** | 0.06 | **0.26 \*** | | | |
| LPC | −0.16 | −0.05 | 0.12 | −0.18 | −0.07 | −0.07 | 0.05 | 0.04 | | |
| C:N | −0.01 | **0.28 \*\*** | **−0.28 \*\*** | **0.28 \*\*** | **0.29 \*\*** | **0.39 \*\*** | **−0.95 \*\*** | 0.03 | −0.05 | |
| N:P | 0.09 | −0.02 | 0.03 | 0.06 | 0.00 | −0.10 | **0.41 \*\*** | 0.09 | **−0.87 \*\*** | **0.38 \*\*** |

Significant correlations (in bold) are denoted by asterisks: \* $p < 0.05$; \*\* $p < 0.01$. Trait abbreviations: LT (leaf thickness); LDMC (leaf dry matter content); LA (leaf area); LMA (leaf mass per unit area); TD (leaf tissue density); WD (wood density); LNC (leaf nitrogen content); LCC (leaf carbon content); LPC (leaf phosphorus content).

### 3.4. CCA Analysis between Functional Traits and Influencing Factors

The interpretation rate of the CCA analysis reached 75.34% on the first axis and 17.29% on the second axis. The correlations between STP and LA, slope and WD, STN and C:N, altitude and TD, and pH and LNC were the greatest, and all were positive. STP and E (elevation) were the most influential factors on plant traits (Figure 3).

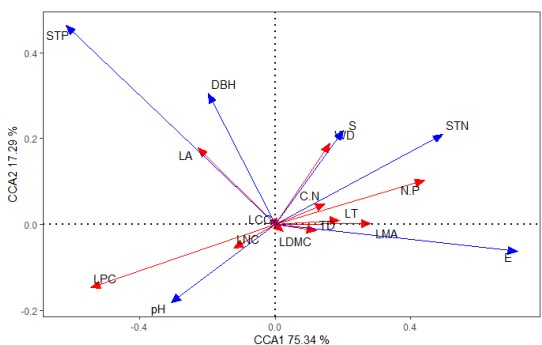

**Figure 3.** Canonical correspondence analysis (CCA) of the functional traits and influencing factors of *L. fragrantissima* var. *lancifolia*. The red solid lines represent functional traits and the blue solid lines represent the influencing factors. The length of the arrow represents the degree of correlation between a certain influencing factor or plant growth index and plants. The angle between the arrows indicates the correlation of the influencing factors. Influencing factors abbreviations: STP (soil total phosphorus); STN (soil total nitrogen); E (elevation); S (slope). Trait abbreviations: LT (leaf thickness); LDMC (leaf dry matter content); LA (leaf area); LMA (leaf mass per unit area); TD (leaf tissue density); WD (wood density); LNC (leaf nitrogen content); LCC (leaf carbon content); LPC (leaf phosphorus content).

## 4. Discussion

### 4.1. Reasons for Variations of Functional Traits of L. fragrantissima var. lancifolia

LT is related to resource availability, solar radiation [31], water storage [32], and $CO_2$ assimilation [33]. It is thicker in a nutrient-poor environment [34,35], which is consistent with the variation patterns of STP. A thicker LT of small shrubs indicates that small shrubs have stronger resistance to water loss and a higher water retention rate [36]. LA is related to the shading degree of the environment, with a high shading degree and large LA to capture more light [37]. On the other hand, when exposed to strong light radiation, plants will reduce LA to avoid water loss through transpiration [38]. Therefore, the LA at low altitudes is larger, but at high altitudes is smaller. Higher LMA indicates the increase in blade structural robustness and is beneficial to slow down leaf senescence, maintain a higher photosynthetic rate and promote dry matter accumulation. LDMC reflects the balance ability of plants between resource acquisition, transformation, and utilization [39,40]. LDMC is typically positively correlated with drought and negatively correlated with temperature [41]. In this study, the LDMC shows a fluctuating trend with altitudes, which may be the result of the comprehensive function of drought degree, less nutrients, and temperature. With the increase in LMA and LDMC of plants, the resistance of water diffusion from leaves to the surface increases, thus reducing the water lost by transpiration and enhancing the adaptability in adverse environments [42]. The increase in WD indicates that plants have experienced the transformation from a resource acquisition strategy to resource conservation strategy. Because *L. fragrantissima* var. *lancifolia* grows under the crown canopy, the light condition changes little, and the range of diameter classes is concentrated, so other traits do not show significant changes with diameter classes. The data of seedlings and old shrubs can be further supplemented.

C:N in leaves reflects the ability of plants to assimilate C by absorbing nutrients, which can reflect the nutrient utilization efficiency of plants to some extent. With the elevation, the C:N increases, indicating that the C utilization efficiency of *L. fragrantissima* var. *lancifolia* in the study plot shows a trend of increasing with elevation. Moreover, the nutrient content for leaves of *L. frangrantissima* var. *lancifolia* at high altitudes is low, which may be due to the strong light at high altitudes, which can appropriately reduce the input of nutrient elements related to photosynthesis in leaves. As we expected, this is consistent with the overall variation patterns of STP and STN [43]. Although the availability of nutrient elements is quite different in different habitats, the N:P of *L. frangrantissima* var. *lancifolia* is all below 12, which indicates that nitrogen in the study plot is relatively scarce, and *L. frangrantissima* var. *lancifolia* is widely limited by nitrogen. Overall, the LPC of *L. fragrantissima* var. *lancifolia* is high, which is related to the shady environment under the forest. It is the choice of evolutionary strategy of *L. fragrantissima* var. *lancifolia* [44]. Compared with high-altitude habitats, low-altitude habitats have a smaller LMA, a higher LNC and LPC, and belong to the type of quick investment return, which is contrary to the research results of Reich et al. [17] and Han et al. [45]. Gullies had a high LNC, probably because it is a valley where rainwater is easy to collect, which promotes the dissolution of STN and makes it easier for plants to absorb it. However, this needs to be further proved by experiments.

### 4.2. Low Variation in Functional Traits in L. fragrantissima var. lancifolia

The low variability in *L. fragrantissima* var. *lancifolia* indicates that Qinling Mountain has good water conditions and a suitable growth environment, which can promote the growth of shrubs in different diameter classes. Moreover, all the shrubs in the different diameter classes were mature shrubs, and the low variation in traits allow plants to devote more resources to blossom, bear fruit, and breed offspring [30]. Because *L. fragrantissima* var. *lancifolia* is clustered, the branches of the trunk root were sometimes collected during sampling time, which leads to great variability in WD. LCC is the basic nutrient component of leaves, and it scarcely varies in leaves among the shrub diameter classes or habitats, indicating that there is a similar photosynthetic efficiency. However, the variation coefficient of LNC is larger than that of LCC, while the variation coefficient of LNC is lower than that

of LPC, which is closely related to the soil nutrient content. The possible reason is that LNC plays the role of LCC in the structural rigidity of leaves in shade-intolerant species [14]. The variation coefficient of LDMC is smaller than that of LMA, which is consistent with Garnier's research results [46].

### 4.3. Leaf Economic Spectrum of L. fragrantissima var. lancifolia

The positive correlation shows that there is a synergistic effect between the two functional traits. The increase or decrease in one element in a plant indicates an increase or decrease in another element, while a negative correlation shows antagonism. The negative relationship between WD and LA can be explained as an adaptation to water limitation. Both smaller leaves and higher WD can improve water-use efficiency [42,47]. The positive correlation between WD and TD is due to the common response to cooler and drier climate and poorer soil environment, which reduces the proportion of cell gaps in leaves [48]. LMA and LDMC were positively correlated. They are both related to net photosynthetic rate, relative growth rate, and net primary productivity at the community level. This result is consistent with the research conclusions reported by Roche [41]. This finding may have implications for the management of forest ecosystem. The negative correlation between LMA and LNC indicates that when plants increase the traits related to leaf structure, they will decrease the traits related to photosynthesis and respiration at the same time [49]. The significant correlation among functional traits of the shrub are consistent with the LES.

### 4.4. The Response of L. fragrantissima var. lancifolia to Various Influencing Factors

Higher STP leads to higher LPC. Phosphorus is important components of carboxylase in photosynthesis. More photosynthetic raw materials need more storage space, so LA is larger. The steep slope has limited water storage capacity, so plants will choose a slow growth strategy, resulting in higher WD. The relationship between STN and C:N verifies the "nutrition luxury hypothesis" [50,51]. This hypothesis suggests that slow-growing species absorb more nutrients than their own demand for growth (luxury consumption) in habitats with poor soil nutrients. They may use their own reserves to support their growth after soil nutrients are depleted [52]. This way of nutrient absorption is conducive to the survival of plants in a highly competitive environment, which indicates that plant species have positive nutrient retention strategies. This leads to a positive correlation between STN and C:N. Higher LD can reduce the physical damage of drought to leaves [53], in order to adapt to the environment with relatively little water at high altitudes. In addition, the study shows that the higher the pH value of the soil, the higher the microbial activity [54] and the stronger the soil respiration [55], and the $CO_2$ flux of soil also increases [56]. An improvement in microbial activity is beneficial to decomposition and the mineralization of soil nitrogen, and thus to the growth of plants.

## 5. Conclusions

We found that the functional traits of *L. fragrantissima* var. *lancifolia* did not change significantly with diameter classes, but changed with habitats, which was affected by the heterogeneity of the habitat. The variation in *L. fragrantissima* var. *lancifolia* was low, and it grew well in the Qinling area. The relationships between the intraspecific traits were consistent with the LES, which increased the basis for the universality of the LES. STP was the major factor affecting the LA of *L. fragrantissima* var. *lancifolia*. The results provided clues for understanding the driving factors of trait variation and are a reference for the introduction and cultivation of *L. fragrantissima* var. *lancifolia*. Future studies should focus on clarifying the responses of the functional traits of *L. fragrantissima* var. *lancifolia* to seasonal variation in climatic factors.

**Author Contributions:** Conceptualization, A.H. and Q.Y.; methodology, A.H.; investigation, A.H. and J.Q.; data curation, A.H. and S.J.; writing—original draft preparation, A.H.; writing—review and editing, R.C. and Q.Y.; supervision, Z.H.; project administration, S.J.; funding acquisition, Z.H. and Q.Y. All authors have read and agreed to the published version of the manuscript.

**Funding:** This study was financially supported by the National Natural Science Foundation of China (32001171, 32001120) and the Fundamental Research Funds for the Central Universities.

**Data Availability Statement:** The data presented in this study are available on request from the corresponding author.

**Conflicts of Interest:** The authors declare no conflict of interest.

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
