# Peer review of "Shrubs Should Be Valued: The Functional Traits of Lonicera fragrantissima var. lancifolia in a Qinling Huangguan Forest Dynamics Plot, China"

_forests, doi:10.3390/f13071147_

Round 1
Reviewer 1 Report
General Comments:
The manuscript described an extensive study of leaf, stem, and soil characteristics for Lonicera frangrantissima var.lancifolia. The species itself is of interest because of its extensive use in horticulture. The paper contains interesting information in regard to how the traits of L. fragrantissima var. lancifolia vary with habitat in the Qinling Mountains of China. Overall, it appears that the methods were appropriate and that some of the results are interesting. However, the writing requires some major editing and there are a few issues with the data presentation that need to be addressed before the manuscript can be published.
One issue with the manuscript was the title begins with "Shrub should not be overlooked:" and yet the species that was studied was referred to as a tree throughout the manuscript. The term "shrub" was primarily used in the Title, Abstract, descriptions of tables, and titles of subsections, but not in the body of the text. It was quite confusing and without a picture of the species, it was hard to know if the authors were studying a shrub or a tree. If the authors were studying a shrub, "tree" should be changed to "shrub" throughout the manuscript. If the authors were studying trees, then the importance of the paper is diminished because the whole premise was that it is important to study functional traits of shrubs.
Another issue is with how small the individual panels are for Figures 2 and 3. It is too difficult to read the figures without zooming in a great deal. Perhaps the panels should be stacked on top of each other to at least increase the height of the individual panels, so that the fonts of the labels can be larger. Further, there needs to be a greater description of the x-axes for both Figure 2 and 3. The units for diameter at breast height (DBH) need to be included in Figure 2. The use of DBH for the figure also leads back to the question as to whether the species was a tree or a shrub. Normally, shrubs have several major branches at breast height. Was a particular branch used to measure DBH? Was there a main branch with many smaller branches? It is not clear. Finally, the actual locations should be used on the x-axes for Figure 3 instead of numbers. It is very hard to look at the figure and know which number corresponds to which habitat without constantly looking at the figure legend.
Continuing with the figures, one has to wonder if most of the panels for Figure 2 are necessary. Only one regression was significant, leaf thickness (LT) v. DBH, but even that had an incredibly low R2 value of 0.05. It is apparent by looking at the panel that DBH has very little influence on LT. Perhaps DBH has little influence on any of the studied traits because Lonicera frangrantissima var.lancifolia plants are shrubs and have multiple major branches. Because of the minimal influence DBH has on the traits studied, I think the results of the regressions (the equations, P-values, and R2) in Figure 2 should be displayed in a table.
Revisiting Figure 3, there is a question as to whether leaf thickness (LT) really increased with elevation. Leaf thickness actually only differs between the two highest elevations and three of the four lowest elevations. Also, after looking at the characteristics of the sites, leaf phosphorous content (LPC) appears to depend on aspect and elevation, because the two sites with the lowest LPC had aspects between 327 and 345 degrees. Perhaps more specific characteristics of each site should be considered when looking at each plant functional trait.
There is a question as to how and why Ward's (the "W" should be capitalized in the Materials and Methods) method was used. It is unclear in the Habitat Classification subsection. Was it used to determine the habitats along with the geography of the sites? If so, then the first sentence should begin with "Because of the differing altitudes, slopes..."
Finally, it would have helped the author's arguments in the Discussion, especially because the authors refer to photosynthesis and exposure, if they presented PAR or solar radiation at the different sites. Having such measurements would have provided further support for the differences in leaf area and leaf thickness with location.
It is believed that if these issues and the specific comments below can be addressed, that the manuscript will be acceptable for publication.
Specific Comments:
1. “Shrub” should be changed to “Shrubs” in the title.
2. In the Abstract, the sentence beginning with “Elevated altitude” should be changed to “Leaf thickness increased with elevation, but leaf phosphorous content decreased with elevation.”
3. In the last sentence of the Abstract, “impact” should be changed to “impactful.”
4. In the fourth line of the first paragraph of the Introduction “could” should be changed to “can.”
5. In the second line of the second paragraph of the Introduction, the comma should be deleted and “and are” should be placed before “systematically.”
6. In the fifth line of the second paragraph of the Introduction, “could” should be deleted and “decreased” should be changed to “decreases.”
7. In the next to last line of the second paragraph of the Introduction, the “s” in “tends” should be deleted.
8. In the third paragraph of the Introduction, “it” is used in lines 11 and 15. Does “it” refer to LES? It is not clear.
9. In the next to last line of the fourth paragraph of the Introduction, “traits of native” should be changed to “traits for native.”
10. In line nine of the fifth paragraph of the Introduction, should “main distribution areas” be changed to “centers of distribution.”
11. In line 10 of the fifth paragraph of the Introduction, the genus name should be spelled out because it is the beginning of the sentence.
12. In the first line of subsection 2.1, “has been” should be changed to “was.”
13. Why were soil samples only taken 10 cm deep into the soil? Was that depth chosen because of root characteristics of the species?
14. In the last line of subsection 2.1, should “drawing” be changed to “creation of figures?”
15. In the third line of subsection 3.1, “(Figure 2)” should be inserted at the end of the sentence if Figure 2 is retained.
16. In the first sentence of subsection 3.3 “but negatively correlation” should be changed to “and correlated negatively.”
17. In reference to the last sentence of subsection 3.3, it is not clear how a trait can be retained, but other cannot.
18. In the first paragraph of subsection 4.1, it is stated that LT is related to CO2 assimilation, but a source for that information was not cited. Also, it should be added that even though LT was significantly related to DBH, it is questionable as to how much DBH contributed to changes in LT.
19. In the first paragraph of subsection 4.1, the sentence that begins with “That the LDMC” should read “LDMC is typically positively…”
20. In the second paragraph of subsection 4.2, the sentence that begins with “Moreover,..” should read “Moreover, the nutrient content for leaves of L. frangrantissima var.lancifolia at high altitudes is low…”
21. The last sentence of the second paragraph of subsection 4.2 needs to be rewritten. I am not sure why the ease of collecting rainwater is related to LNC. Is it due to the solubility of nitrogen? Erosion?
22. In the first line of subsection 4.2, it is not clear what the small variation is related to. Plant size? Or just overall?
23. In line 11 of subsection 4.2, is photosynthetic “intensity” supposed to be “efficiency” or “flux?”
24. In the second line of subsection 4.3, the authors need to define what the “two factors” are. It is unclear.
25. In the tenth line of subsection 4.3, it is stated that “LM and LDMC were positively correlated with net photosynthetic rate,” but photosynthesis was never measured. Do the authors mean LM and LDMC “are” positively correlated to net photosynthetic rate in the literature? Photosynthesis was not measured in the study.
26. In subsection 4.4, the authors need to better explain the “nutrition luxury hypothesis and why plants would take on that strategy in nutrient-poor soil.
27. In Conclusions, “obviously” should be deleted from line 3.
Reviewer 2 Report
Dear authors,
This is an interesting and well written manuscript. I really enjoyed reading this research and the dataset is good.
1. All key elements are present.
2. The title clearly describes the article.
3. The abstract content clearly reflects the entire content of the article.
4. In the introduction paragraph the authors clearly estate the problem investigated. Also, the purpose of the study is specified.
5. The authors accurately explain the field experiments.
6. Data are well presented
Reviewer 3 Report
This manuscript presents what could potentially have been nice results. However, I have several doubts about the case study, and I am dissatisfied with the theoretical framing. I sometimes felt as if you were “trying something”, but were not sure what this “something” is.
One important comment: please add line numbers, unless instructed otherwise. It is difficult to follow and comment on a manuscript when I cannot refer to a specific line.
It seems that the word “tree” is often used instead of “shrub”.
Title
“Shrub should not be overlooked” – this pre-assumes that shrubs are overlooked. Is this really the case? Even is the term “overlook” is justified, I think it is a little over-provocative for a title.
I would also refrain from using the word “adaptability” in the title. This is a study of intra-specific plant functional trait variations and of plant trait syndrome spectrum. Adaptability is an interpretation more than an actual conclusion. Even in the third sentence in the body text, you note that functional traits can reflect adaptability, which is a long way from indicating adaptability. From an eco-physiological standpoint, I doubt if there is any test of adaptability in this study.
Abstract
The latter part of the abstract includes too many variables and acronyms, making it hard to follow and sound too technical. I suggest instead focusing on the more general trends, even if this entails presenting processes results (i.e., being more Discussion than Results). I would use the last sentence of the abstract as the foundation: describe how functional trait syndromes vary along the elevation and STP gradients.
Introduction
I found the explanation of functional traits a little shallow and missing key terms such as “response” (to environmental conditions) and “effect” (on plants and ecosystems). I also dislike the statement that functional traits are affected by plant size. While this is sometimes expressed in the literature, I think it is more appropriate to consider plant size (or allometry as a whole as) a functional trait. That is, a plant’s functional traits vary throughout its lifetime (reflected in part by its size), and this variation is by itself a part of the “plant’s economic spectrum” (in lack of a better term in the literature). In the second paragraph, there is a mixture of both sources of variation: phenological and ecotypical.
Page 2:
“functional traits [form] leaf economic spectrum” – even traits that are not related to the leaf? Most of the first paragraph on this page summarizes known basic trends of the LES, which I believe are well known among the potential readership of this manuscript.
“On a global scale, functional traits are mainly affected by climate” – you should state that earlier in the manuscript. I also think that “abiotic factors, mainly climate” would be a more cautious statement (especially considering the size of your study site).
Third paragraph: The move towards the specific case study is not smooth. There should be at least one sentence discussing the general case to which the case study belong. The description and justification of the study area and species is also lacking. First, nothing is said about L. fragrantissima’s abundance and role in local ecosystems. Second, while you mention several dominant tree species, you do not mention whether or how L. fragrantissima may respond to the dominant tree(s) under which it grows.
Methods
The difference between diameter classes and life history stages is unclear. Are the latter not simply categories of the former? When reading the Results section, it feels as if the same trends are shown twice, once with a continuous x-axis and then with a categorical one.
The use of linear regressions only is dubious, since other models may potentially be better (although Fig. 2 does not indicate this clearly). I think you should at least justify why you pre-assumed linearity, or state that other models had lower fit.
Results
Fig. 3: Please write the habitat type on the x-axis, instead of referring us to the legend.
Discussion
This section felt overall too long to read, and not focused enough.
The first sentence is an unproven pre-assumption, which should be a proposition based on the discussion (and hence concluding it). It seems as if you have pre-determined your conclusions!
Section 4.2: Does this not mean that you chose the wrong case study to test how environmental conditions affect shrub intraspecific functional trait variations?
Page 10, first sentence: Which two factors?
Round 2
Reviewer 3 Report
I must admit that I am somewhat disappointed from this revised manuscript. It seems that the authors mostly dealt with cosmetics and minor revisions, but seem to have neglected some important comments about the theoretical framing (e.g., providing a makeshift definition of functional traits instead of using the one used in the literature), teleology (insisting to write about adaptability and pre-assuming it, although it is not studied in any way and the fact that pre-assuming it leads the discussion instead of being its conclusion), and some questions regarding the quality of the case study.
Under these circumstances, I cannot conclude that the manuscript has really been revised, and all my previous comments remain unanswered.
